# Sensitivity Analysis of RV Reducer Rotation Error Based on Deep Gaussian Processes

**DOI:** 10.3390/s23073579

**Published:** 2023-03-29

**Authors:** Shousong Jin, Shulong Shang, Suqi Jiang, Mengyi Cao, Yaliang Wang

**Affiliations:** College of Mechanical Engineering, Zhejiang University of Technology, Hangzhou 310032, China

**Keywords:** RV reducer, rotation error, deep Gaussian processes, sensitivity analysis

## Abstract

The rotation error is the most important quality characteristic index of a rotate vector (RV) reducer, and it is difficult to accurately optimize the design of a RV reducer, such as the Taguchi type, due to the many factors affecting the rotation error and the serious coupling effect among the factors. This paper analyzes the RV reducer rotation error and each factor based on the deep Gaussian processes (DeepGP) model and Sobol sensitivity analysis(SA) method. Firstly, using the optimal Latin hypercube sampling (OLHS) approach and the DeepGP model, a high-precision regression prediction model of the rotation error and each affecting factor was created. On the basis of the prediction model, the Sobol method was used to conduct a global SA of the factors influencing the rotation error and to compare the coupling relationship between the factors. The results show that the OLHS method and the DeepGP model are suitable for predicting the rotation error in this paper, and the accuracy of the prediction model constructed based on both of them is as high as 95%. The rotation error mainly depends on the influencing factors in the second stage cycloidal pinwheel drive part. The primary involute planetary part and planetary output carrier’s rotation error factors have little effect. The coupling effects between the matching clearance between the pin gear and needle gear hole (δJ) and the circular position error of the needle gear hole (δt) is noticeably stronger.

## 1. Introduction

The term “rotate vector (RV) reducer” typically refers to a two-stage reduction mechanism consisting of the first-stage involute planetary gear transmission and the second-stage cycloidal pin-wheel transmission. This mechanism has the advantages of a compact structure, a high transmission accuracy and a large reduction ratio, and it is frequently used in high precision equipment in Computerized Numerical Control (CNC) machine tools, aerospace and industrial robotics industries [1,2]. The most crucial performance metric for an RV reducer is the positioning transmission precision, which is often quantified by the output rotation error. The rotation error is determined by the precision of the parts processing and the assembly of the transmission chain from the input to the output of the RV reducer, as well as the state of motion and force; even under the same operating conditions, the quality characteristics of each link in the transmission chain do not have the same degree of influence on the RV reducer’s output rotation error. It is challenging to accurately optimize the design of RV reducers, such as the Taguchi type, that will handle the problem satisfactorily because of the transmission chain’s influence from a variety of parameters and the presence of random coupling effects. A sensitivity analysis (SA) of a precise RV reducer’s rotation error is urgently required.

Scholars both at home and abroad are currently conducting extensive research on rotation error analysis. YANG et al. provided an analytical error modeling approach for the RV reducer’s over-constrained structure and demonstrated the link between the original error and the transmission precision [3]. Hidaka et al. used the equivalent spring approach to model the exact system, and they qualitatively examined the transmission accuracy of the entire RV reducer [4,5,6]. Using the dynamic sub-structure method, HE and SHAN developed the transmission error dynamics model and mathematical equations for the RV-40E reducer for robots [7]. CAO et al. created an equivalent model of the transmission error of the RV reducer and solved the model’s parameters using the empirical stiffness formula to determine the theoretical transmission error of the reducer [8]. By analyzing the structural characteristics of the RV reducer, the parts processing and assembly process, and the influence of the manufacturing errors on the clearance, CHU et al. proposed a selective assembly method to make the clearance of the RV reducer meet the requirements [9]. LIU analyzed the transmission accuracy of the reducer from multiple angles, such as manufacturing and assembly errors and external working conditions, using theoretical analysis, numerical simulation, dynamics, and transmission accuracy tests [10]. Meng et al. provided transmission error models for positive and negative transmission by analyzing the static analysis of the actual meshing process between the gear pairs while taking friction into account [11]. The accuracy of the RV reducer’s equivalency error model significantly increased when Liu et al. introduced an RV reducer transmission error modeling and optimization approach and used the particle swarm algorithm to refine the empirical parameters of the identification model [12]. Based on the Adams software and virtual prototype technology, Tong created many virtual prototypes of the RV reducer and investigated the effects of the load, part elastic deformation, and component manufacturing error on the transmission error [13]. JIN et al. proposed a new method for predicting the rotation error based on improved grey wolf optimal support vector regression. This method overcomes the shortcomings of the previous rotation error research methods, such as their time-consuming nature and low computational precision [14]. On the basis of error sensitivity analysis, Hu et al. suggested an elastic transmission error compensation approach for a rotating vector to boost its transmission accuracy [15]. Wu et al. used the dynamics analysis program ADAMS to create a virtual prototype of the RV reducer employed in the robot and tested the transmission accuracy of the Japan RV-40E reduction using the Grating method [16].

The studies mentioned above provide many valuable insights for the analysis and evaluation of the RV reducer rotation error, but the majority of them are based on qualitative analysis results, which have some limitations and do not provide an accurate analysis of the RV reducer rotation error. For example: the calculation process of the pure geometric method is relatively complicated and does not take into account the actual manufacturing error of the parts. Due to a large amount of simplification in the modelling process, the equivalent spring method has a large gap between the final modelling result and the actual situation, and the model construction of the virtual prototype analysis method is complicated and time-consuming. In order to address the limitations of the traditional modelling and analysis methods of the rotation error, this paper proposes a novel modelling method for predicting the rotation error of a RV reducer based on deep Gaussian processes (DeepGP) and conducts sensitivity analysis on each factor influencing the rotation error.

To begin with, the optimal Latin hypercube sampling (OLHS) method is combined with a DeepGP model to develop a high-precision rotation error prediction model. The model’s accuracy and efficiency are validated by comparing it to the spring equivalence method and the virtual prototype. Then, using the variance-based Sobol method, a global sensitivity analysis of the factors influencing the rotation error was performed based on the prediction model to identify a number of factors that had a substantial impact on the rotation error. Lastly, quantitative analysis of the influence of the coupling effect between the factors on the rotation error will provide reference significance for RV reducer quality improvement, cost reduction, and other optimization designs.

## 2. Analysis of the Factors Influencing the Rotation Error of RV-40E Reducer

### 2.1. RV Reducer Transmission Principle and Structural Composition Analysis

An RV reducer is a two-stage transmission mechanism compounded by the first-stage involute planetary gear transmission and the second-stage cycloidal pin-wheel transmission, which has many different composition structures and transmission forms and can be applied to different occasions. A typical RV-40E reducer structure is shown in Figure 1a below, including the input shaft with a sun gear, planetary gear, crankshaft, cycloidal pinwheel, needle tooth shell, planet carrier, needle gear, and other parts. The RV reducer transmission principle as shown in Figure 1b. Through the involute transmission, the input shaft with a sun gear transfers the power to the planetary wheel to complete the first stage of deceleration; the crankshaft and the planetary wheel are fixedly connected at the same speed, the crankshaft drives the cycloid wheel to turn around the needle tooth shell through a slewing bearing, and the rotation of the cycloid gear transmits the planet carrier and flange plate. The planet carrier and flange plate are bolted together at the same speed and act as the output mechanism to complete the second stage deceleration.

The rotation error is the difference between the theoretical output rotation angle and the actual rotation angle; these are calculated by the following formula:(1)$$\varphi_{er}=\varphi_{in}÷i-\varphi_{out}$$
where $\varphi_{er}$ is the rotation error; $\varphi_{in}$ is the input angle of input shaft; $\varphi_{out}$ is the actual angle of the output shaft; i is the transmission ratio. The main structural composition parameters of the RV-40E reducer are shown in Table 1.

### 2.2. Analysis of Factors Influencing the Rotation Error of RV Reducer

The rotation error is a major quality performance metric used to evaluate the performance of the speed reducers. The influencing factors might be classified as static or dynamic. Assembly errors, fit clearance between pieces, and machining faults are examples of static variables. Despite the fact that there are many static elements that affect the rotation error, just one region of the needle tooth shell has more than ten different types of mistakes. If every static piece is evaluated, the work would be vast and unrealistic. Based on a study of the reducer’s transmission chain and actual production research, 15 frequently static influencing elements were selected from the transmission chain of the RV reduction at all levels, and refer to [17] to create the parameter ranges for each factor, as shown in Table 2. Factors 1–3 are generated by the primary involute planetary gear drive, factors 4–12 are generated by the second-stage cycloidal pin-wheel drive, and factors 13–15 are generated by the planetary output carrier. The dynamic factor is primarily the influence of the elastic deformation of each part on the rotation error of the reducer when the reducer is loaded, and when the reducer is running, the dynamic factor consists primarily of the load and the speed. As the purpose of this research paper is to determine the factors influencing the high sensitivity of the RV reducer to the rotation error in order to facilitate the Taguchi-type optimized design of the RV reducer, the factors influencing this sensitivity will be identified. The dynamic influence factor is an external influence factor of the speed reducer rotation error that is not currently considered in this paper.

The aforementioned static influencing factors combine to generate the RV reducer rotation error, which can be represented as follows using the fuzzy comprehensive evaluation mathematical method:(2)$$\varphi_{er}=\omega_{\Delta f_{a}}\delta_{\Delta f_{a}}+\omega_{\Delta F_{r}}\delta_{\Delta F_{r}}+\omega_{\Delta C_{k}}\delta_{\Delta C_{k}}+\omega_{\Delta r_{rp}}\delta_{\Delta r_{rp}}+\omega_{\delta \Delta r_{p}}\delta_{\delta \Delta r_{p}}+\omega_{\sigma }\delta_{\sigma }\;+\omega_{\delta r_{p}}\delta_{\delta r_{p}}+\omega_{\delta r_{rp}}\delta_{\delta r_{rp}}+\omega_{\delta J}\delta_{\delta J}+\omega_{t}\delta_{t}+\omega_{\delta a}\delta_{\delta a}+\omega_{\Delta r}\delta_{\Delta r}\;+\omega_{Eh}\delta_{Eh}+\omega_{\Delta Fa}\delta_{\Delta Fa}+\omega_{\Delta_{2}}\delta_{\Delta_{2}}$$
where $\varphi_{er}$ is the rotation error of the RV reducer, $\delta_{i}$ is the rotation error of each influencing factor, and $\omega_{i}$ is the weight of each factor. The weights represent the overall degree of influence of each influencing factor on the rotation error of the reducer [18].

### 2.3. RV Reducer Rotation Error Sample Data Acquisition

In this study, the factor-error data were obtained directly from the RV reducer production and processing facility to ensure the practicality of the rotation error prediction model and sensitivity analysis results. Each reducer’s rotation error is measured using a dedicated RV reducer rotation error test bench, and the deviation of each influencing factor is measured using the double cylindric rod measuring method, the common normal line method, and other technologies. Figure 2 depicts the schematic diagram of the rotation error test bench. The servo motor can be set to rotate at different speeds, the magnetic powder brake can be used to apply different loads. Two circular encoders record the angular signals at the input and output of the reducer; using the phase difference information of the angular signals, calculate the rotation error of the reducer.

When testing each reducer, the test bench is configured with the same speed and load, the rotation error data of each RV reducer in the stable operation is recorded, and the average value is considered the rotation error value. Adjust the servo motor speed to 1200 rpm, set the load bit to 200 N·m, record the rotation error after the reducer has stabilized, and select at random the test curve chart for three reducers, as depicted in Figure 3. The sample point of the factor-rotation error is the sum of the value of the rotation error received from the test and the deviation of each influencing factor.

## 3. Construction of Prediction Model of RV Reducer Rotation Error Based on the DeepGP Model

Due to the multiplicity of the components and their interactions, the regression of the reducer’s rotation error exhibits high-dimensional nonlinearity. Surrogate models, which have the advantages of simple model development, quick processing, and high prediction accuracy, are now frequently used to solve nonlinear problems. Traditional surrogate models, such as Kriging [19] and the Gaussian process (GP) [20], are frequently constrained by the network structure and kernel functions, resulting in poor fitting accuracy. The DeepGP model stacks multiple Gaussian process regression models, and by combining probabilistic models and deep learning, it retains the excellent performance of Gaussian process regression while having more powerful mapping capabilities in fitting high-dimensional problems [21], and we decided to build a rotation error prediction model based on the DeepGP model in this paper.

There are three essential steps in building the RV reduction rotation error prediction model:Determine the range of values for each rotation error-influencing factor based on the actual production research, and then use the OLHS technique to collect the sample points for the prediction model’s training set.Construct a rotation error prediction model using the DeepGP model and determining the structure and parameters of the model.Use a validation set to evaluate the prediction model’s accuracy in order to make it easier to conduct a sensitivity analysis of each individual influencing element in the following section.

### 3.1. Sample Point Extraction Based on the OLHS Method

During the construction phase of the prediction model, the distribution of the sample points in the modeling space and the total number of sample points have a significant impact on the model’s performance. An adequate number of uniformly distributed sample points can significantly reduce the computation costs while accurately reflecting the change trend and information of the real model in space. In all other respects, the proxy model created using these sample points can accurately and credibly reflect the sample space [22].

The Latin hypercube sampling (LHS) method, which is based on the concept of probabilistic stratification, divides the probability distribution space of the test factors into N nonoverlapping subregions based on the range of values for each influencing factor, and then conducts independent equal-probability sampling in each subregion. This method is more capable of filling empty spaces than the orthogonal test method. Due to the numerous factors affecting the rotation error in this paper, if 100 sample points are to be collected, the sample space must be divided into 100^15^ sub-regions, which significantly increases the computer’s workload by excessively subdividing the spatial sample area. Moreover, there may be a part of the design space missing due to too many areas, so the LHS method is obviously not suitable for the task of sample point sampling in this paper. Based on the LHS approach, the OLHS method can ensure that all design points are uniformly distributed across the design space by evenly and orthogonally distributing them in the design space region of the test factors [23].

The parameter range of each rotation error influence factor was determined in Table 2, and 200 sample points of the influencing factor data were extracted from each parameter range using the OLHS method. The following are the specific sampling steps:1.Each dimensional axis is subdivided into 10 equal intervals, thereby dividing the sample space range into 10^15^ sub-regions with dimensions based on the 15 variables that affect the rotation error of the reducer, such as the $\Delta r_{rp}$.2.Obtaining 200 sub-regions of an optimal Latin hypercube sampling matrix with uniform distribution based on the space-filling optimality criterion, space-filling optimality criterion is shown in the following equation:


(3)
$$\overset{n}{\underset{i=1}{\sum }}\overset{n}{\underset{j=1}{\sum }}d_{ij}^{2}=min\overset{n}{\underset{i=1}{\sum }}\overset{n}{\underset{j=1}{\sum }}\overset{n}{\underset{l=1}{\sum }}{\left(x_{i}^{\left(l\right)}-x_{j}^{\left(l\right)}\right)}^{2}\;$$


In this equation, $x_{i}$ and $x_{j}$ indicate any two points, $d_{ii}$ indicates the distance between any two points, and l indicates the dimension of the sample points, and this equation is used for the OLHS matrix calculation.

3.Following that, a sample point is selected within each of the extracted sub-regions and each parameter sampling, point is then used as the coordinate component of each dimension of the sampling point.

Using the sample point’s coordinate set, each sample point is located in the sample database of the rotation error. Due to the discrete nature of the database, some sample points cannot be located. Instead, sample points near the spatial locations of these coordinate points are chosen. The final extracted partial data of the factor-rotation error sample points are shown in Table 3, which uses millimeters(mm) as the unit of measurement for each influencing factor. These data comprise the training set for the prediction model. Concurrently, 200 sample points were selected at random from the database to serve as a validation set for testing the accuracy of the prediction model developed in the subsequent paper.

### 3.2. Process of Building Rotation Error Prediction Model

#### 3.2.1. Deep Gaussian Processes Model

The DeepGP employs the Gaussian process and deep learning as a hierarchical network to address the nonlinear relationships between the nodes in the hierarchy [24]. Layering numerous single-layer Gaussian processes on top of one another results in a deep network structure. These layers are then linked together. The DeepGP model’s organizational structure is depicted in Figure 4.

The DeepGP model can be expressed as the following function:(4)$$y=f^{Zn}\left(f^{Z\left(n-1\right)}\left(\ldots \ldots f^{x}\left(x\right)\right)\right)+\varepsilon$$
where x is the input layer, y is the output layer, $Z_{i}$ is the hidden layer, and ε is the noise. A Gaussian process with only input and output layers and no hidden layers is standard.

The information transfer rules for the deep Gaussian process model are as follows:(5)$$Z_{1}=f^{x}\left(x\right)+\varepsilon$$
(6)$$Z_{2}=f^{Z_{1}}\left(Z_{1}\right)+\varepsilon_{1}^{Z_{1}}$$
⋯
(7)$$Z_{n}=f^{Z_{\left(n-1\right)}}\left(Z_{\left(n-1\right)}\right)+\varepsilon_{\left(n-1\right)}^{Z_{\left(n-1\right)}}$$
(8)$$y=f^{Z_{n}}\left(Z_{n}\right)+\varepsilon_{n}^{Z_{n}}$$

The layer’s nodes represent the input values of the network’s lower layer and the output values of the network’s higher layer. A Gaussian process is utilized to map the two layers of neurons. The layers are connected by a kernel function that automatically modifies the connections between a large number of nodes and determines the ideal model parameters while assuming different weights for each conceivable dimension. According to Table 4, the most popular varieties of kernels are as follows.

DeepGP is no longer a straightforward Gaussian process due to its multi-layered network structure and complex mapping relationships. Bayesian inference methods are no longer applicable to DeepGP model parameter estimation. Due to the complexity and high dimensionality of the parameter space and the space where the hidden variables are located, variational inference is frequently employed in deterministic approximation inference to compute the model’s hidden variables [25]. Figure 5 depicts the training optimization flowchart of the DeepGP model. The training of the model consists primarily of variational inference within the DeepGP layers, variational inference between the DeepGP layers, and DeepGP parameter optimization, and the specific steps are as follows:

Step1: The sample data are separated into training and test sets, and the number of input variables is determined.

Step2: Initialize the model parameters and variational parameters of the DeepGP model.

Step3: Input sample training data to train the model.

Step4: Variational inference within the DeepGP model layer: using a sparse variational inference method to derive an initial variational lower bound on the original probability space.

Step5: Variational inference between DeepGP model layers: maximizing the log-edge likelihood of the target data and obtaining the best approximation of log(p(y/X)).

Step6: DeepGP model parameter optimization: after determining the induced variables, the model structure and model parameters are determined using the conjugate gradient algorithm with the lower bound of the variance as the objective function.

Step7: Lastly, using the validation data to assess the constructed model’s accuracy.

The mean absolute percentage error (MAPE), the coefficient of determination (R^2^), the mean absolute error (MAE), and the root mean squared error (RMSE) are also used to evaluate the local and global accuracy of the developed model. The formulas for calculating the four performance indicators are provided below:(9)$$\{\begin{array}{c} MAPE=\frac{\sum_{i=1}^{n}\left|{\hat{y}}_{i}-y_{i}\right|}{y_{i}}\;\\ R^{2}=1-\frac{\sum_{i=1}^{n}{\left({\hat{y}}_{i}-y_{i}\right)}^{2}}{\sum_{i=1}^{n}{\left({\hat{y}}_{i}-\bar{y_{i}}\right)}^{2}}\\ MAE=\frac{\sum_{i=1}^{n}\left|{\hat{y}}_{i}-y_{i}\right|}{n}\;\\ RMSE=\sqrt{\frac{\sum_{i=1}^{n}{\left(y_{i}-{\hat{y}}_{i}\right)}^{2}}{n}} \end{array}$$
where n denotes the number of sample points, $y_{i}$ denotes the measured value of sample points, ${\hat{y}}_{i}$ denotes the predicted value of sample points, and $\bar{y_{i}}\;$ denotes the average value of sample points.

#### 3.2.2. Construction of RV Reducer Rotation Error Prediction Model

The hyperparameters of the DeepGP model consist of the number of hidden layers, the number of nodes within each layer, and the type of kernel between each layer. These hyperparameters are critical for the model’s accuracy and effectiveness. Theoretically, the more hidden layers there are in a deep network, the smaller the model’s regression error will be. However, if there are too many layers, the model’s training time will be excessively long, it will be overly intricate, and it will have overfitting issues. The literature [26] suggests that DeepGP models with two or three hidden layers perform well in terms of prediction. In this work, the hidden layers of the model are chosen as two layers to maintain the prediction speed while reducing the model complexity.

Erickson et al. [27] concluded that GPy: a Gaussian Process framework written in Python outperforms other packages. As a result, this study uses GPy packages to build a DeepGP model. The number of nodes in the input layer corresponds to the number of variables that influence the rotation error of the reducer. The output layer variable is the rotation error, and the output layer contains a total of one layer. Therefore, the model-based DeepGP model and the input X and output layers Y can be expressed as follows:(10)$$X=\left[x_{1},x_{2},\ldots ,x_{15}\right]=\left[\Delta f_{a},\Delta F_{r},\ldots ,\Delta F_{a},\Delta_{2}\right]$$
(11)$$Y=\varphi_{er}$$
(12)$$y=f^{Z2}\left(f^{Z1}(f^{x}\left(x\right)\right)+\varepsilon$$

As the rotation error increases gradually as the error value of each influencing factor increases, the rotation error is linear with each influencing factor as a whole. As a result, the linear kernel was selected as the kernel function between the input layer and the initial hidden layer; The RBF kernel is selected as the kernel function between the first and second hidden layers and between the second hidden layer and the output layer due to the severe coupling effect between the influencing factors.

In this paper, the ideal number of nodes in the two hidden layers is determined by the grey wolf optimization (GWO) algorithm [28]. The training set is utilized to construct the prediction model, and the fitness function is the R^2^ of the model’s ability to predict the validation set collected above. The wolf count (P) of the population is set to 30, and the number of iterations (N) is set to 100. It is determined through an iterative search that the prediction model with the highest R^2^ value is constructed when the number of nodes in the first hidden layer is six and the number of nodes in the second hidden layer is three for the validation set, as well as the prediction accuracy of this model. As a result, Figure 6 depicts the structure of the RV reducer rotation error prediction model developed using the DeepGP model.

#### 3.2.3. Accuracy Check of Rotation Error Prediction Model

Using the 200 validation sample points collected above, the accuracy of the prediction model developed for this study is evaluated. MAPE, R^2^, MAE, and RMSE are chosen as the performance metrics to evaluate the model’s regional and overall accuracy. In order to reduce the chance factor in the test data, the validation set was evenly divided into two for the model regression testing. Table 5 shows the test results for the two validation sets. In this study, the prediction model has a prediction accuracy of more than 95% for both validation sets. The prediction model in this paper has a low prediction error and good fitting accuracy.

The rotation error dynamics model proposed in the literature [7] based on the equivalent spring error model and the virtual prototype constructed in the literature [13] were selected for comparison in order to validate the applicability and superiority of the rotation error prediction model constructed based on the DeepGP model proposed in this paper. The equivalent spring schematic model of the RV reducer constructed using the dynamic substructure method is depicted in Figure 7a, below, where *K*i represents the stiffness of each part. The virtual prototype of the RV reducer constructed in the Adams dynamics software is depicted in Figure 7b below.

As the virtual prototype simulation is time-consuming and inefficient, this paper arbitrarily selects 40 points from the validation set to compare the three models’ accuracy verification. The test results of the three models are depicted in Table 6 and Figure 8 below.

It can be seen from the Table 6 that the R^2^ value of the prediction model presented in this article reaches 96%. The model presented in this paper has stable mapping and high prediction accuracy, whereas the R^2^ value of the other two models is just over 75%. Figure 8 shows that the values predicted in this article are more in line with the true value, while the rotation error predicted by the other two models deviates from the measured values and are, in most cases, smaller than the measured values. Due to the fact that this paper is based on the actual production and processing plant data, the other two models may not account for the influence of the friction force and disregard the influence of certain components, such as the bearings, in the modeling process.

According to the model evaluation results, the samples collected using the OLHS approach were uniform and representative, which matched the criteria for the sampling points required by the prediction model and could guarantee the model’s construction accuracy. This paper extracts the data directly from the actual production and processing plant to assure the applicability and precision of its analysis results. The RV reducer rotation error modelling method presented in this paper, which is based on the DeepGP model, is faster and more precise than the traditional modelling techniques. The mapping stability of the prediction model constructed in this paper paves the way for a later sensitivity analysis of the factors affecting rotation error.

## 4. Sensitivity Analysis of Factors Influencing the Rotation Error of RV Reducers Based on Sobol Method

The examination of the rotation error prediction model reveals that it is stable in mapping and has a high level of prediction accuracy, meeting the requirements of sensitivity analysis (SA). As a result, the SA can use the newly developed rotation error prediction model to reduce the computation costs and improve the analysis efficiency.

### 4.1. Sobol Sensitivity Analysis Method

SA focuses on the relative magnitude of the effect of the model input on the output uncertainty, which identifies the important design variables and is crucial for the optimal design of engineering products. SA is divided into local SA and global SA. In the current process, global sensitivity is frequently used to analyze the issue. The common sensitivity analysis techniques used today include the Fourier amplitude sensitivity test (FAST) [29], Morris [30], one-factor-at-a-time methods with Latin-hypercube sampling (LH-OAT) [31], Generalized Likelihood Uncertainty Estimation (GLUE) [32], and the Sobol method [33]. According to the analysis principles, the sensitivity analysis methods can be divided into qualitative and quantitative methods. Table 7 lists the characteristics of the different sensitivity analysis methods.

Comparing the characteristics of the aforementioned sensitivity analysis methods, the variance-based Sobol method considers the influence of the parameter interactions on the results and can not only solve the local and global sensitivity indexes of each uncertainty variable, but can also quantify the influence of the parameter interactions on the results. In this paper, the Sobol method is employed to quantify the sensitivity indexes of each factor influencing the rotation error and to examine the influence of the interaction and coupling between the factors on the rotation error.

For the multivariate function y=g(x), it can be decomposed into the following summation form:(13)$$g\left(x\right)=g_{0}+\overset{k}{\underset{i=1}{\sum }}g_{i}\left(x_{i}\right)+\underset{1\leq i\leq i\leq k}{\sum }g_{ij}\left(x_{i},x_{j}\right)+\ldots +g_{1,2,\ldots ,k}\left(x_{1},x_{2},\ldots ,x_{k}\right)$$

The right term of the above equation is expressed in the form of a multiple integral as follows:(14)$$g_{0}=\int^{\;}\ldots \int^{\;}g\left(x\right)dx=\int^{\;}\ldots \int^{\;}g\left(x_{1},\ldots ,x_{k}\right)dx_{1}\ldots dx_{k}$$
(15)$$g_{i}\left(x_{i}\right)=\int^{\;}\ldots \int^{\;}g\left(x\right)dx_{-i}-g_{0}$$
(16)$$g_{ij}\left(x_{i},x_{j}\right)=\int^{\;}\ldots \int^{\;}g\left(x\right)dx_{-\left(ij\right)}-g_{0}-g_{i}\left(x_{i}\right)-g_{j}\left(x_{j}\right)$$
where $dx_{-i}$ denotes the integral of variables other than $x_{i}$, $dx_{-\left(ij\right)}$ denotes the integral of variables other than $x_{i}$ and $x_{j}$, the remaining terms of Equation (13) can be derived to find.

The variance V and bias a of y can be expressed as:(17)$$V=\int^{\;}\ldots \int^{\;}g^{2}\left(x_{1},\ldots ,x_{k}\right)dx_{1}\ldots dx_{k}-g_{0}^{2}$$
(18)$$\;V_{i_{1}\ldots i_{s}}=V\left[g_{i_{1}\ldots i_{s}}\right]=\int^{\;}\ldots \int^{\;}g_{i_{1}\ldots i_{s}}^{2}\left(x_{i_{1}},\ldots ,x_{i_{s}}\right)d_{x_{i_{1}}}\ldots d_{x_{i_{s}}}$$
where $1\leq i_{1}<\ldots <i_{s}\leq k$, $V_{i_{1}\ldots i_{s}}/V$ denotes the sensitivity of $\left\{i_{1},\ldots ,i_{s}\right\}$. Sensitivity indicators for different order variables can be obtained by adjusting the number of $\left\{i_{1},\ldots ,i_{s}\right\}$. Among them, the first-order Sobol sensitivity indicator $S_{i}$ and the global sensitivity indicator $S_{T_{i}}$ are commonly used.

The first-order indices are also known as the main effects, and the corresponding expression is shown in the equation below:(19)$$S_{i}=V_{i}/V$$

Other higher-order Sobol sensitivity indices can also be obtained based on the bias variance, and higher-order Sobol indices indicate the cross-correlation between two or more factors. First-order and higher-order indices have the following properties:(20)$$\sum_{i=1}^{k}S_{i}+\sum_{i<j}S_{ij}+\ldots +S_{ij\ldots k}=1$$

The total-order indices are also known as the total effects, and can be expressed as:(21)$$S_{T_{i}}=1-S_{{-}_{i}}$$
where $S_{{-}_{i}}$ denotes the sum of all $S_{ij\ldots k}$ excluding variable $x_{i}$.

$S_{i}$ denotes the first-order effect of the output parameter $x_{i}$ alone on the output result, and the $S_{i}$ does not account for covariate cross-talk. $S_{T_{i}}$ denotes the sum of the first-order and higher-order effects of the input parameter $x_{i}$ on the output results, and it evaluates the parameter $x_{i}$’s overall effect on the model output. When $S_{T_{i}}=0$, the variable $x_{i}$ has no effect on the model output. In this paper, we will calculate each sensitivity index using a Monte Carlo sampling method based on variance [34].

### 4.2. Sensitivity Analysis Process for Rotation Error Influencing Factors Based Sobol Method

The Python package SALib is used as the primary research tool in this study, and the Sobol method is employed to conduct a sensitivity analysis of the variables influencing the RV reducer’s rotation error. As a result of the small sample size used in this study, the results of the SA using these sample points alone may not be accurate enough. The inhomogeneity of the sample location space will lead to numerous inaccuracies if a large number of samples are collected at random. As a result, the previously mentioned rotation error prediction model is extracted first, followed by the use of SALib’s proprietary Saltelli sampler to create uniform samples, and finally the Sobol method to calculate the SA for each influencing component. Each test was repeated three times under the same conditions to ensure that the sampling had no effect on the SA results, and the average of all the values was used to determine each SA’s final result.

#### 4.2.1. Results of Total Effects of Factors Influencing the Rotation Error of RV Reducer

The SA method was built in Python software. Firstly, the prediction model of the reducer rotation error constructed above was imported; then, each influencing variable and its value range were determined, and the SA problem model of this paper was defined; finally, 2400 uniform samples were generated by the Saltelli sampler for the quantitative analysis of 15 influencing factors of the reducer rotation error.

To begin, the total effects of each rotation error influence factor are analyzed, and the total effects indicators of the reducer rotation error influencing factors are displayed in Table 8, where a and the other identifiers represent each component influencing the rotation error. As shown in the table, the total effect of δt(j) is 0.568193, making it the factor with the highest index. δt is the most sensitive to the rotation error’s effects. The effects of $\Delta r_{rp}$, $\Delta r_{p}$, σ, $\delta r_{p}$, $\delta r_{rp}$, δJ, δt, Δr are more than ten times as much as the remaining influencing factors’ effects. Analyzing the structure and transmission principle of the reducer reveals that its first-stage planetary gear reduction mechanism is located far from the output end, and its influence on the rotation error has been greatly reduced due to the error it produces after the second stage reduction transmission. The combined force of two 180° out of phase cycloid wheels is applied to the planetary frame. The impact of the planetary output carrier on the RV reducer’s rotation error is significantly mitigated [35]. The rotation error of the RV reducer can be attributed to the error affecting the factor of the secondary cycloidal pin-wheel transmission mechanism.

#### 4.2.2. Analysis of the Coupling Effect of Factors Influencing the Rotation Error of RV Reducer

The involute planetary transmission component and the planetary output carrier have a very small and negligible impact on the rotation error, as demonstrated by the above study; thus, we will ignore these aspects for the time being. Figure 9 depicts the primary influences and overall effects of each contributing element on the rotation error of the RV reducer. Where each influence component, denoted by a data label, is the same as in Table 8. As depicted in the figure, $\delta r_{rp}$(h) and δt(j) have higher sensitivity indices and their impact on the rotation error is marginally greater. Their overall and primary effects are comparable. The remaining factors, such as $\Delta r_{rp}\;$(d) and Δr (l), vary substantially in terms of their main effects and total effects, despite having relatively smaller sensitivity indicators. As these factors interact significantly with other factors, it is vital to study each factor’s coupling relationship. The second-order sensitivity index of the factor influencing the rotation error is then analyzed.

Figure 10 shows a heat map of the second-order sensitivity index between the variables affecting the rotation error of the RV reducer, with the labels of the rows and columns representing the same error factors as in Table 8. An increase in the thermogram readings or a darkening of the color of the two elements indicates that the coupling between the two components has a greater influence on the reducer’s rotation error. As shown in the picture, the second-order sensitivity index between δJ(i) and δt(j) is 0.029, and the second-order sensitivity index between δJ(i) and $\delta r_{rp}$(h) is 0.025. These two coupling effects are noticeably stronger. The second-order sensitivity index between $\Delta r_{rp}$(d) and $\Delta r_{p}$(e) is 0.0077, and the coupling effects between $\Delta r_{rp}$(d) and $\Delta r_{p}$(e) have less of an impact on the rotation error. As the second-order sensitivity index between the other two components was either zero or negative, none of the coupling effects were significant. By comparing the first-order, second-order, and total effects of the factors influencing the rotation error, we can conclude that the total effects of each factor is produced by the combination of the main effects and the coupling between itself and the other influences.

#### 4.2.3. Comparison of Sensitivity Analysis Results

We compare the Sobol method with the established Taylor midvalue theorem sensitivity analysis method [13] to more thoroughly assess the Sobol method’s accuracy. In the literature [13], the equivalence error method based on the principle of the equivalent spring method was first used to derive the value of the rotation error caused by each factor of the rotation error, followed by the Taylor midvalue theorem to obtain the sensitivity coefficient for each factor, and finally the sensitivity index for each factor. The results of the sensitivity comparison of some of the factors influencing the rotation errors calculated by the Taylor and Sobol methodologies are shown in Table 9.

According to Table 9, when comparing the normalized sensitivity index of each factor deviation derived by the equivalent error method with the Taylor midvalue theorem and the sensitivity index calculated in this paper, the calculation results of the two methods have little difference, the sensitivity index of each factor deviation is in the same order, and the sensitivity qualitative analysis of each factor deviation is the same. The approach proposed in the literature [13] does not consider the effect of the coupling effects between factors and is more of a qualitative analysis. In contrast, this paper analyzes the combined effect of each factor on the rotation error, solves the sensitivity index of each factor quantitatively, and examines how the coupling between the factors affects the rotation error.

As a result, the rotation error prediction model and the sensitivity analysis method presented in this paper are correct.

## 5. Conclusions

Using the DeepGP model and the Sobol sensitivity approach, this paper investigates the sensitivity of each contributing component of the RV reducer rotation error. The preceding analysis leads to the following conclusions:Using the OLHS method, actual production facilities were sampled to ensure the accuracy of the created prediction models, as well as the dependability and applicability of the SA results. Using the Deep GP model, a high-precision prediction model for the rotation error of each RV reducer was developed, and the accuracy was compared with the equivalent method and virtual prototypes to demonstrate the validity and accuracy of the prediction model, thereby establishing the conditions for the sensitivity analysis of the RV reducer’s rotation error.On the basis of the prediction model, a global sensitivity analysis of the variables affecting the rotation error of the RV reducer was conducted using the Sobol method. The primary cause of the reducer’s rotation error is the second-stage cycloidal pin-wheel transmission mechanism, with the influence of the planetary gear transmission and the planetary output carrier being small and negligible.The second-order sensitivity index is used to evaluate how multiple influencing elements interact to affect the rotation error of the reducer. There is significantly more coupling between δJ and δt, as well as between $\delta r_{rp}\;$ and δJ. The coupling effect between other variables has little effect on the rotation error. The total order sensitivity index of each factor is calculated by adding its main effects and the coupling of those effects to other influences. The results of this paper’s sensitivity analysis were contrasted with the results of Taylor’s midvalue theorem to confirm the originality and precision of the analysis results.

## Figures and Tables

**Figure 1 sensors-23-03579-f001:**
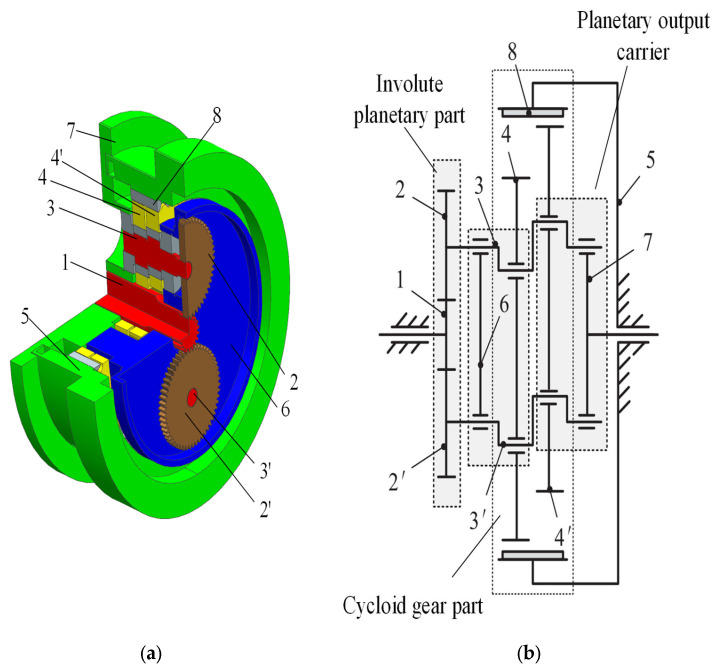
Schematic diagram of RV-40E reducer structure principle. (**a**) The structure of RV reducer; (**b**) Description of what is contained in the second panel. 1. Input shaft with sun gear; 2 (2′). Planetary gear; 3 (3′). Crankshaft; 4 (4′). Cycloid gear; 5. Needle tooth shell; 6. Planet carrier; 7. Flange plate; 8. Needle gear.

**Figure 2 sensors-23-03579-f002:**
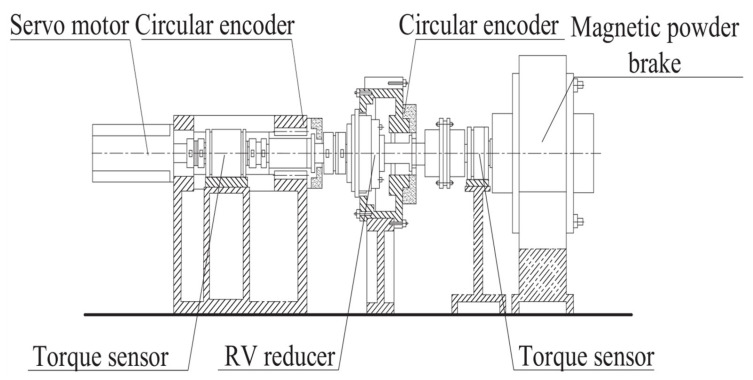
Schematic diagram of RV reducer rotation error test bench.

**Figure 3 sensors-23-03579-f003:**
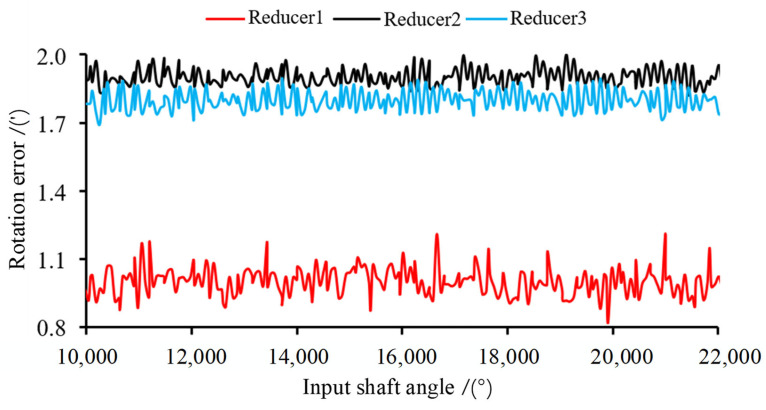
Test curve chart for reducers.

**Figure 4 sensors-23-03579-f004:**
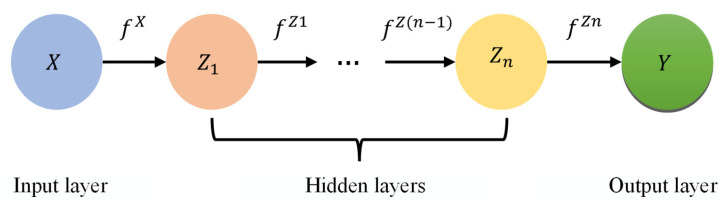
The structure of the DeepGP model.

**Figure 5 sensors-23-03579-f005:**
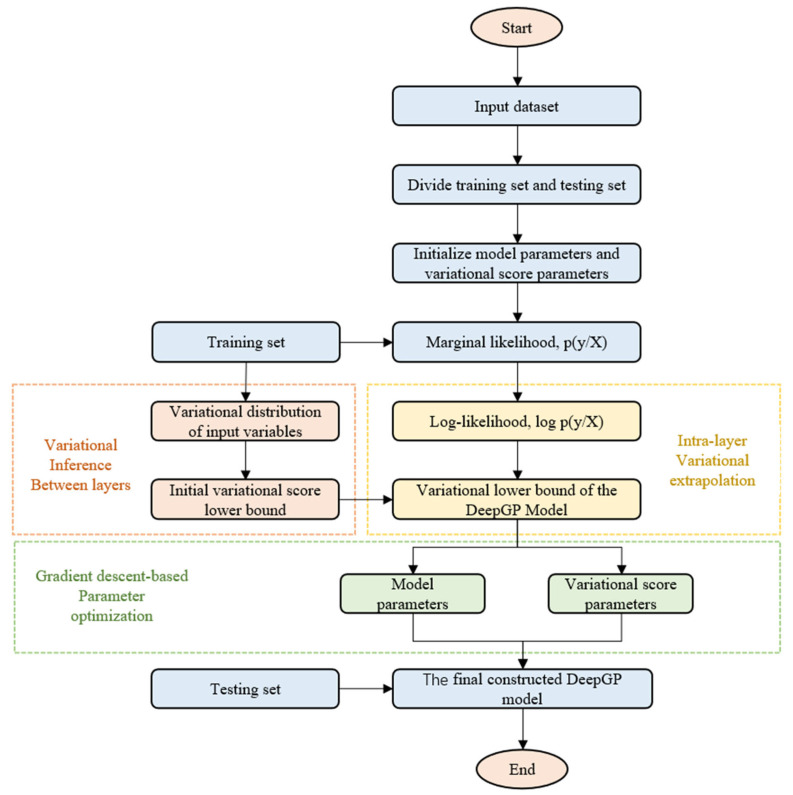
DeepGP model training flow chart.

**Figure 6 sensors-23-03579-f006:**
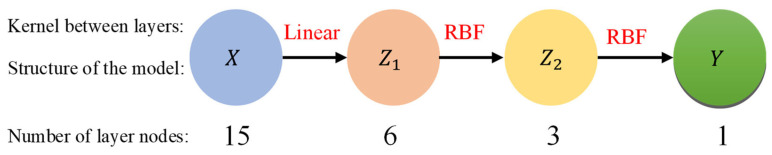
Structure diagram of the rotation error prediction model.

**Figure 7 sensors-23-03579-f007:**
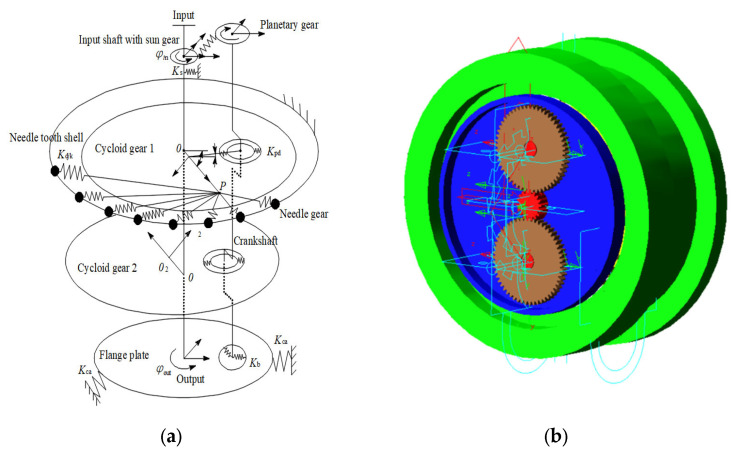
Two model diagrams of RV-40E Reducer. (**a**) Equivalent spring error model of RV-40E reducer. (**b**) Virtual prototype of RV-40E reducer.

**Figure 8 sensors-23-03579-f008:**
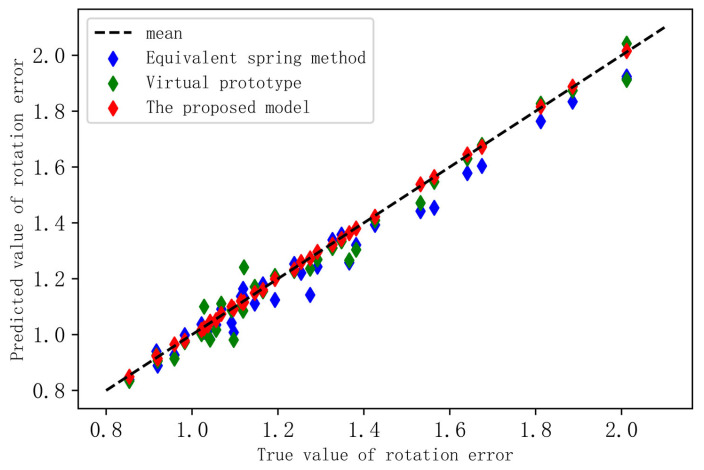
Test results of this model and two other models.

**Figure 9 sensors-23-03579-f009:**
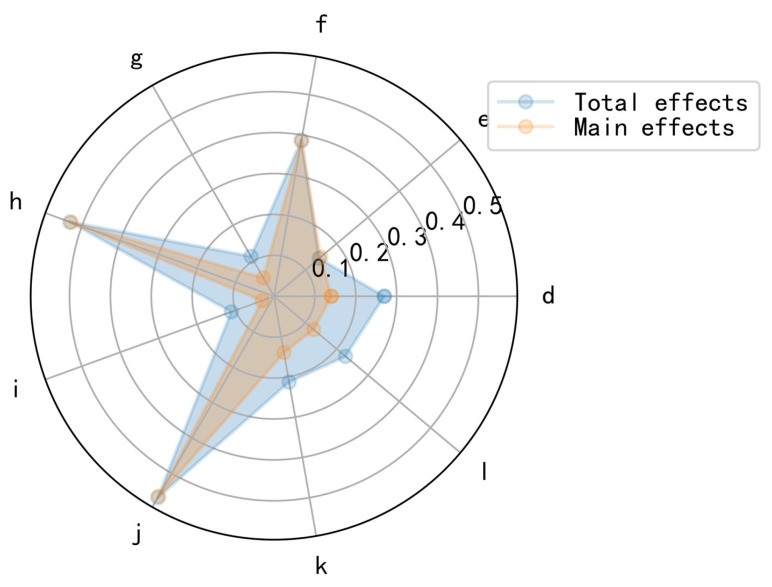
The main effects and total effects of each influencing factor of RV reducer rotation error.

**Figure 10 sensors-23-03579-f010:**
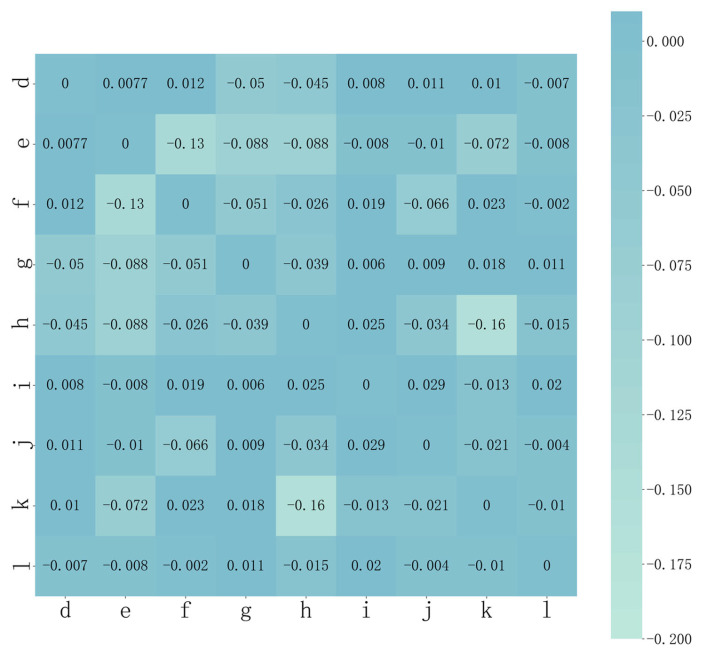
Heat map of second-order sensitivity of RV reducer rotation error.

**Table 1 sensors-23-03579-t001:** The main parameter of RV-40E reducer.

Parameters	Value
Number of teeth of sun gear and planetary gear	18/54
Number of teeth of cycloid gear (z_c_) and needle gear	39/40
Eccentricity of crankshaft (a) (mm)	1.5
Circle center radius of needle tooth shell (*δ*r_p_) (mm)	82
Radius of needle gear (*δ*r_rp_) (mm)	4
Total transmission ratio (i)	121
Short amplitude factor (k_1_)	0.73
Input power	1.05 (kW)
Nominal torque	377 (N∙m)

**Table 2 sensors-23-03579-t002:** RV reducer rotation error static influence factors and parameters to take the value range.

Sources of Influencing Factors	No.	Influencing Factor of Rotation Error	Code Name	The Range/mm
The primary involute planetary part	1	Center distance error of input shaft and planetary gear	$\Delta f_{a}$	0.15~0.50
	2	Radial error between gear and ring	$\Delta F_{r}$	0.20~0.30
	3	Fit clearance of gear and shaft	$\Delta C_{k}$	0.001~0.003
The second stage cycloidal pinwheel part	4	Amount of equidistant modification of cycloid gear	$\Delta r_{rp}$	0.016~0.048
	5	The amount of radial-moving modification of cycloid gear	$\Delta r_{p}$	0.026~0.078
	6	Clearance of inside hole of cycloid gear and slewing bearing	σ	0~0.01
	7	Radius error of needle tooth center circle	$\delta r_{p}$	0.005~0.015
	8	Needle gear radius error	$\delta r_{rp}$	0.01~0.03
	9	Matching clearance between the pin gear and needle gear hole	δJ	0.001~0.003
	10	Circular position error of the needle gear hole	δt	0~0.01
	11	Eccentricity error of crankshaft	δa	0.01~0.03
	12	Clearance of crankshaft and slewing bearing	Δr	0~0.002
Planetary output carrier	13	Eccentricity error of planet carrier	Eh	0.005~0.015
	14	Carrier wheel crankshaft center distance error	ΔFa	0~0.01
	15	Bearing hole phase error of planet carrier	$\Delta_{2}$	0.03~0.05

**Table 3 sensors-23-03579-t003:** The set for training rotation error prediction model.

Sample Point	$\left(\Delta f_{a},\;\;\Delta F_{r},\;\;\Delta C_{k},\;\;\Delta r_{rp},\;\;\Delta r_{p},\;\;\sigma ,\;\;r_{p},\;\;r_{rp},\;\;\delta J,\;\;\delta t,\;a,\;\;\Delta r,\;Eh,\;\;\Delta Fa,\;\;\Delta_{2}\right)$	Rotation Error/’
1	(0.037, 0.200, 0.001, 0.047, 0.044, 0.009, 82.015, 4.027, 0.001, 0.001, 1.519, 0.001, 0.011, 0.006, 0.037)	0.059
2	(0.164, 0.297, 0.002, 0.038, 0.072, 0.003, 82.011, 4.016, 0.002, 0.005, 1.525, 0.001, 0.011, 0.001, 0.041)	1.238
3	(0.325, 0.277, 0.001, 0.022, 0.047, 0.001, 82.013, 4.030, 0.001, 0.004, 1.528, 0.002, 0.010, 0.008, 0.034)	1.538
⋯	⋯	⋯
100	(0.164, 0.224, 0.003, 0.030, 0.076, 0.002, 82.009, 4.026, 0.002, 0.003, 1.516, 0.002, 0.008, 0.005, 0.041)	1.483
101	(0.420, 0.258, 0.002, 0.028, 0.063, 0.005, 82.012, 4.024, 0.002, 0.002, 1.529, 0.001, 0.008, 0.002, 0.044)	2.161
⋯	⋯	⋯
199	(0.436, 0.242, 0.003, 0.036, 0.042, 0.004, 82.015, 4.018, 0.003, 0.001, 1.515, 0.001, 0.011, 0.03, 0.047)	1.217
200	(0.286, 0.233, 0.002, 0.044, 0.068, 0.002, 82.008, 4.026, 0.001, 0.003, 1.510, 0, 0.007, 0.008, 0.050)	1.040

**Table 4 sensors-23-03579-t004:** The commonly used kernel functions.

Kernel Function	Expression
The Linear kernel	$k\left(x,x^{\prime };\theta \right)=xx^{\prime }$
The absolute-exponential kernel (RBF)	$k\left(x,x^{\prime };\theta \right)=\sigma^{2}\left(-\frac{1}{2}{\left[\frac{\left(x-x^{\prime }\right)}{l}\right]}^{2}\right)$
The absolute-exponential kernel	$k\left(x,x^{\prime };\theta \right)=\sigma^{2}exp\left(\frac{\left|x-x^{\prime }\right|}{l}\right)$
The Periodic covariance kernel function (PER)	$k\left(r\right)=\sigma^{2}exp\left[-\frac{2sin^{2}\left(\pi \left|x-x^{\prime }\right|/p\right)}{l^{2}}\right]$

Where, l denotes the characteristic length scale. p denotes the period.

**Table 5 sensors-23-03579-t005:** Results of model regression accuracy test.

Performance Metrics	Validation Set 1	Validation Set 2
MAPE	0.05533	0.07584
R^2^	0.97871	0.95913
MAE	0.06059	0.07085
RMSE	0.07053	0.08143

**Table 6 sensors-23-03579-t006:** Comparison of regression accuracy of three models.

Performance Metrics	The Proposed Method	Equivalent Spring Method	Virtual Prototype
MAPE	0.06533	0.17326	0.20461
R^2^	0.96846	0.78542	0.76941
MAE	0.08059	0.13212	0.12941
RMSE	0.07053	0.16819	0.20851

**Table 7 sensors-23-03579-t007:** Characteristics of sensitivity methods.

Sensitivity Analysis Methods		Characteristics of Sensitivity Methods
Local sensitivity analysis	FAST	Simple operation, ignores parameter interactions, low model applicability.
Global sensitivity analysis	Morris	Compared the output results of adjacent parameters in the parameter space, inefficient.
	LH-OAT	Although the benefits of both the random one-factor-at-a-time and LHS sampling techniques are taken into account, the computer program is complex.
	GLUE	Combining the benefits of Rivest-Shamir-Adleman techniques and fuzzy mathematics to rank sensitivities as scatter plots.
	Sobol	The sensitivity indices are solved using the Monte Carlo sampling technique, which can distinguish independently between parameter-independent and parameter-interacting sensitivities.

**Table 8 sensors-23-03579-t008:** The total effects indicators of the reducer rotation error influencing factors.

Label	Influencing Factor of Rotation Error	Total Effects
a	$\Delta f_{a}$	0.012414
b	$\Delta F_{r}$	0.025427
c	$\Delta C_{k}$	0.007129
d	$\Delta r_{rp}$	0.269588
e	$\Delta r_{p}$	0.144105
f	σ	0.387645
g	$\delta r_{p}$	0.112615
h	$\delta r_{rp}$	0.531029
i	δJ	0.111341
j	δt	0.568193
k	δa	0.021295
l	Δr	0.228339
m	Eh	0.014437
n	ΔFa	0.012762
o	$\Delta_{2}$	0.009721

**Table 9 sensors-23-03579-t009:** Comparison of the sensitivity indexes of the factors influencing the rotation error obtained by the two methods.

Infuencing Factor of Ratation Error	Taylor Sensitivity Method			Sobol
	Rotation Error Caused by Each Factor	Sensitivity Coefficient	Sensitivity Index	Total Effects
$\Delta f_{a}$	$\frac{\Delta f_{a}\times 180\times 60}{\pi \times i\times r_{1}}$	-	0.025	0.0124
$\Delta F_{r}$	$\frac{\Delta F_{r}\times 180\times 60}{\pi \times i\times r_{1}}$	-	0.025	0.0254
$\Delta r_{rp}$	$\frac{2\times \Delta r_{rp}}{a\times Z_{c}}$	1	1	0.269588
$\Delta r_{p}$	$\frac{2\times \Delta r_{p}\times \sqrt{1-k_{1}^{2}}}{a\times z_{c}}$	$\sqrt{1-k_{1}^{2}}$	0.68	0.144105
$\delta r_{p}$	$\frac{2\times \delta r_{p}\times \sqrt{1-k_{1}^{2}}}{a\times z_{c}}$	$\sqrt{1-k_{1}^{2}}$	0.68	0.112615
$\delta r_{rp}$	$\frac{-2\times \delta r_{rp}}{a\times z_{c}}$	1	1	0.5312
δJ	$\frac{\delta J}{a\times z_{c}}$	0.5	0.5	0.1111
δt	$\frac{k_{1}\times \delta J}{a\times z_{c}}$	$k_{1}$	0.73	0.56819
Δr	$\frac{180\times 60\times \Delta r}{\pi \times a}$	$\frac{ez_{c}}{2a}$	0.679487	0.228339

Where $r_{1}$ denotes the radius of the input shaft with sun gear.

## Data Availability

The data presented in this study are available on request from the corresponding author. The data are not publicly available, because they involve corporate data privacy.

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
