# Peer review of "Sensitivity Analysis of RV Reducer Rotation Error Based on Deep Gaussian Processes"

_sensors, 2023, doi:10.3390/s23073579_

Round 1

Reviewer 1 Report

Dear authors,

This is one very interesting paper in RV reducers area with good introduction, literature review, results and conclusions. For RV reducers and their modern application (industrial robots, CNC machines,…) their precision and repeatability are very important working characteristics. From these reason rotation error analysis is very important.

Here are some comments:

-       -  In literature review please add the paper: A.D. Pham, H.J. Ahn: High Precision Reducers for Industrial Robots Driving 4th Industrial Revolution: State of Arts, Analysis, Design, Performance Evaluation and Perspective, International Journal of Precision Engineering and Manufacturing-Green Technology, Vol. 5, No. 4, pp. 519-533, 2018. This is one very interesting paper about high precision reducers and their applications at industrial robots.

-        -  What is the input power of the RV-40E reducer and its nominal torque?

-  How many reducers from production line (approximately) you analysed for several months of work?

-       -  Figure 1: Instead of 1. Input shaft write 1. Input shaft with sun gear.

 - Figure 1: Please in describe insert and components 2`, 3` and 4`. For example: 2 (2`). Planetary gear; 3(3`). Crankshaft; 4(4`). Cycloid gear.

-        - Table 1: Instead of The number of teeth of input shaft and planetary gear write The number of teeth of sun gear and planetary gear.

-        - Table 2: In column „The Range/mm“ give the source of this data.

-  Equation (2) is cut out on the left side.

 -  From the aspect of design and production, the cycloid gear is the most complex element of the RV-40E reducer. How do you explain the results that influencing factors of rotation error which are connected with cycloid gear do not have a bigger influence?

Reviewer 2 Report

It is interesting to predict the RV reducer rotation error by combining the OLHS method and the DeepGP model. Comparing this proposed method and those rotation error models using the equivalent spring method and virtual prototypes would be better.

Reviewer 3 Report

The authors investigated the sensitivity of each contributing component of RV reducer rotation error using the DeepGP model and the Sobol' sensitivity approach. The paper is well-written and interesting to read, however, I see the following significant issues that should be resolved before publishing this paper:

1- The authors should state the originality of this study clearly.

2- The authors should add a flow chart of Deep Gaussian Processes Model to explain more.

3- Different methods for sensitivity analysis should be discussed.

4- In this manuscript, there is no comparison between different methods. The authors should show the accuracy of the proposed method 

5- Authors found this conclusion "The primary cause of the reducer's rotation error is the second-stage cycloidal pin-wheel transmission mechanism, with the influence of the planetary gear transmission and the planetary output carrier being small and negligible." Can we find the same result using any method apart from sensitivity analysis?

6- Can we use the high-precision prediction model that was built in this study for more complex problems?

7- The resolutions of Figures 7 and 8 are not enough for publication. 

Round 2

Reviewer 3 Report

The authors have addressed all my comments for this paper and answered the technical questions I have about this method. The paper has been significantly improved after revision. The revised manuscript is ready for publication.